# Admission Lactate Concentration, Base Excess, and Alactic Base Excess Predict the 28-Day Inward Mortality in Shock Patients

**DOI:** 10.3390/jcm11206125

**Published:** 2022-10-18

**Authors:** Piotr Smuszkiewicz, Natalia Jawień, Jakub Szrama, Marta Lubarska, Krzysztof Kusza, Przemysław Guzik

**Affiliations:** 1Department of Anesthesiology, Intensive Therapy and Pain Management, Poznan University of Medical Sciences, 60-355 Poznan, Poland; 2Department of Cardiology—Intensive Therapy, Poznan University of Medical Sciences, 60-355 Poznan, Poland

**Keywords:** base excess, alactic base excess, hyperlactatemia, mortality, shock

## Abstract

Base excess (BE) and lactate concentration may predict mortality in critically ill patients. However, the predictive values of alactic BE (aBE; the sum of BE and lactate), or a combination of BE and lactate are unknown. The study aimed to investigate whether BE, lactate, and aBE measured on admission to ICU may predict the 28-day mortality for patients undergoing any form of shock. In 143 consecutive adults, arterial BE, lactate, and aBE were measured upon ICU admission. Receiver Operating Curve (ROC) characteristics and Cox proportional hazard regression models (adjusted to age, gender, forms of shock, and presence of severe renal failure) were then used to investigate any association between these parameters and 28-day mortality. aBE < −3.63 mmol/L was found to be associated with a hazard ratio of 3.19 (HR; 95% confidence interval (CI): 1.62–6.27) for mortality. Risk of death was higher for BE < −9.5 mmol/L (HR: 4.22; 95% CI: 2.21–8.05), particularly at lactate concentrations > 4.5 mmol/L (HR: 4.62; 95% CI: 2.56–8.33). A 15.71% mortality rate was found for the combined condition of BE > cut-off and lactate < cut-off. When BE was below but lactate above their respective cut-offs, the mortality rate increased to 78.91%. The Cox regression model demonstrated that the predictive values of BE and lactate were mutually independent and additive. The 28-day mortality in shock patients admitted to ICU can be predicted by aBE, but BE and lactate deliver greater prognostic value, particularly when combined. The clinical value of our findings deserves further prospective evaluation.

## 1. Introduction

Shock is a life-threatening haemodynamic emergency associated with tissue and organ hypoperfusion, leading to severe metabolic changes at the cellular level. Low oxygen delivery, high oxygen consumption, or inadequate utilisation can prompt various metabolic derangements, thus resulting in an acid–base imbalance with or without hyperlactataemia.

Patients in shock are routinely admitted to an intensive care unit (ICU). Mortality is usually high in such patients but varies depending on clinical status, for example, due to metabolic abnormalities. Identifying the most at-risk patients is necessary for successful treatment to prevent further deterioration, to reverse metabolic derangements, and to reduce mortality. In addition to standard risk scores, e.g., Sequential Organ Failure Assessment (SOFA) [1], various parameters have been evaluated in critically ill individuals for assessing mortality risk.

Predictive value has been demonstrated for acid–base balance parameters or lactate concentration and its kinetics in ICU patients [2,3,4,5,6,7,8], including those with septic shock, unstable haemodynamics, cardiac arrest, urgency laparotomy, and patients requiring ECMO support during transportation [9,10,11,12,13,14,15].

Lactate metabolism in shock patients is very complex. Traditionally, hyperlactataemia is interpreted as an indicator of anaerobic metabolism due to tissue hypoxia [16,17,18]. It may also reflect abnormal oxygen extraction and utilisation, effects of prolonged excess of catecholamines, presence of veno-arterial shunts in peripheral tissues [19,20,21,22,23,24], mitochondrial dysfunction [25], liver dysfunction impairing lactate clearance [26], and thiamine depletion due to shock-induced upregulated metabolic processes [27].

Hyperlactataemia may be accompanied by metabolic acidosis or as a separate phenomenon not affecting the acid–base status. In 2019, Gattinoni et al. [28] proposed the term alactic base excess (aBE) to distinguish between metabolic acidosis caused by lactate accumulation and that caused by increased amounts of non-lactate fixed acids (unmeasured strong anions) not eliminated through the lungs, e.g., phosphate and sulphate acids. Conceptually, aBE is a sum of negative values of BE and lactate. Gattinoni et al. [28] studied aBE in septic patients, concluding that it possibly reflects the impact of renal dysfunction on plasma lactate concentration and acid–base balance. They have also demonstrated that increased plasma lactate indicates sepsis severity and that aBE may be used to estimate renal capability to control acid–base balance.

Nevertheless, the predictive value of aBE has not yet been investigated in either septic or non-septic ICU patients. In this study, we aim to investigate a potential prognostic value of aBE and its contributors, i.e., BE and lactate concentration to predicting 28-day mortality in shock patients admitted to ICU. Additionally, we examine whether the prognostic values of BE and lactate are simultaneously independent and additive.

## 2. Materials and Methods

### 2.1. Study Design and Population

It is a retrospective observational study using clinical data of patients hospitalized in a teaching hospital’s mixed medical–surgical ICU. The study included one hundred forty-three consecutive adult patients (minimum age 18 years) with any form of shock, unconscious at admission, requiring intravenous norepinephrine and mechanical ventilation. All were new patients not hospitalized before at any other ICU or another hospital for the current clinical state. The only exclusion criterion was moribund status. Patients were managed with standard procedures, including treatment of the underlying cause, lung protective mechanical ventilation, hemodynamic monitoring, fluid resuscitation, supply of vasoactive agents, and other clinically justified interventions. In the case of septic shock, patients were treated according to the guidelines of the Surviving Sepsis Campaign, including the appropriate administration of antibiotics [29]. The scientific purpose of this study did not affect any clinical decisions or the duration of inward stay. According to the enrolment criteria, patients were unconscious at the time of admission. Thus, the Bioethics Committee at the Poznan University of Medical Sciences in Poland waived the requirement to obtain informed consent and approved the study protocol. All patients were given unique codes to keep their anonymity in the database. No names or contact details were circulating among researchers involved in the study. In this way, patients’ anonymity was preserved. All methods were carried out following relevant guidelines and regulations.

#### Data Collection

Patient information on demographics, Acute Physiology and Chronic Health Evaluation II’s (APACHE II) [30], and SOFA score [1] were collected. The results of standard clinical and laboratory parameters were also collected, including arterial and central venous blood gas analysis measured at the bedside within the first 5–10 min after admission to the ICU (Radiometer ABL 90 Flex Plus). Standard BE and aBE per patient were calculated according to Gattinoni [28] as follows:standard BE (mmol/L) = [HCO_3_ (mmol/L) − 24.8 (mmol/L)] + 16.2 mmol/L × (pH − 7.4)
alactic BE (mmol/L) = standard Base Excess (mmol/L) + lactate (mmol/L)

### 2.2. Data Coding and Statistical Analysis

Qualitative data were represented as overall numbers and percentages or by subgroups of survivors and non-survivors. All continuous data were summarised using either mean values with standard deviation (SD), or medians with the 25th and 75th percentiles (IQR) due to normal or non-Gaussian distributions (the D’Agostino-Pearson normality test).

Total mortality during the 28-day hospitalisation at the ICU was used as a primary end-point. Patients discharged earlier were treated as censored without further follow-up, whereas patients who required extended ICU stay were labelled as alive. Those who were re-admitted to the ICU were not considered as separate hospitalisations. The date of death, discharge from ICU, or end of follow-up was recorded for each patient. All patients were divided into categories of those who survived or died during the 28-day ICU stay for comparisons of (1) continuous data (with either the unpaired t-test or Mann–Whitney test, as appropriate); (2) binomial data (with the Fisher exact test). The Receiver Operating Curve (ROC) characteristics test was used to analyse the association of BE, lactate concentration, and aBE with 28-day mortality, using optimal cut-off values for these parameters determined by the Youden criterion. The Cox proportional hazard models were adjusted to the patient’s age, gender, type of shock (four categories: septic, cardiac, hypovolemic, and other), and the presence of estimated glomerular filtration rate (eGFR) <30 mL/min/1.73 m^2^. The results were then shown as Hazard Ratios (HR) with their 95% Confidence Interval (95% CI). Different Cox Hazard models were built based only on cut-off values of BE, lactate concentration, or aBE. Additional models for BE and lactate concentration cut-off values were built according to the following criteria:category 0—if BE was above (less severe or no acidaemia) and lactate concentration was below (non-severe or no hyperlactataemia) their respective cut-off values, i.e., no patient had severe acidaemia and hyperlactataemia;category 1—if either BE was below or lactate concentration was above their respective cut-off values, i.e., patients with either severe acidaemia or severe hyperlactataemia;category 2—if both BE was below and lactate concentration was above their respective cut-off values, i.e., patients with coexisting severe acidosis and hyperlactataemia.

Finally, both BE and lactate concentration values were entered into the same Cox Hazard model to investigate whether they have independent and additive predictive values.

Mutual associations between indices of renal function, i.e., between creatinine concentration and estimated glomerular filtration rate (eGFR), or between BE, aBE, and lactate concentration, were analysed using the Spearman correlation. The correlation between aBE and either lactate concentration or BE was not analysed since aBE is mathematically related to both.

Statistical differences with *p* < 0.05 were considered to be significant. All statistical analyses were performed using the MedCalc Statistical Software version 19.1 (MedCalc Software bv, Ostend, Belgium) or PQStat version 1.8.2.202 (PQStat Software, Poznan, Poland).

## 3. Results

Table 1 summarises the comparison of clinical characteristics related to the main hypotheses between survivors and non-survivors. The clinical characteristics of all studied patients are shown in Appendix A for qualitative and Appendix A for quantitative data. The comparisons of other clinical characteristics between survivors and non-survivors are presented in Appendix A for the qualitative and Appendix A for the quantitative data.

Non-survivors were older by six years; had higher APACHE II and SOFA scores, 10 points and 3 points, respectively; and required additional vasoactive drugs more frequently than survivors. Hypovolemic shock was less frequent but SaO_2_ < 94% was present in every second patient from non-survivors. Their pH, standard HCO_3_, BE, and aBE were found to be significantly lower (reduced to approximately −7 mmol/L and −4.5 mmol/L for BE and aBE, respectively), whereas lactate concentration was higher (>2.5 mmol/L). Non-survivors had significantly more frequent reduced bicarbonates, acidaemia, and hyperlactatemia. Median values of systolic and diastolic BP were also lower in non-survivors than survivors at the admission to ICU.

AUCs from the ROC analysis were observed to be largest for BE and smallest for aBE, with the AUC for BE being larger than that of aBE (*p* = 0.0049). No significant differences in AUCs were observed when comparing lactate concentration with either BE (*p* = 0.4946) or aBE (*p* = 0.4355). ROC analysis results with AUCs and identified cut-off values of BE, lactic concentration, and aBE are shown in Figure 1.

Table 2 summarises mortality rates for specific subgroups of patients stratified according to the defined cut-offs for BE; lactate concentration; aBE; and categories 0, 1, and 2, reflecting various combinations of the cut-offs for BE and lactates. The 28-day ICU mortality in shock patients was found to be lowest for category 0 (15.71%) and highest for category 2 (78.94%).

In unadjusted and adjusted Cox regression models, patients with BE < −9.5 mmol/L, lactate concentration > 4.5 mmol/L, or aBE < −3.63 mmol/L had significantly increased risk of premature death during the 28-day ICU stay (Table 3).

In these models, a change in one category from 0 to 1, or from 1 to 2 was also significantly associated with mortality. Figure 2 shows survival curves derived from Cox proportional hazard models for patients stratified by either the cut-off values for BE; lactate concentration; aBE; or categories 0, 1, or 2 (i.e., three combinations of BE and lactate cut-offs).

As BE and lactate concentrations (dichotomised according to their cut-off values) have prognostic values both as single covariates or combined into categories, it was also investigated whether they were simultaneously independent of one another and additive. The final adjusted Cox regression model (chi^2^ = 50.13; *p* < 0.0001) confirmed these assumptions with a hazard ratio of 2.54 (95% CI 1.25–5.15; *p* = 0.0100) for BE < −9.5 mmol/L and 3.02 (95% CI 1.58–5.78; *p* = 0.0008) for aBE.

From the nonparametric Spearman correlation, it was found that BE values and lactate concentrations were significantly and negatively correlated (rho = −0.58; *p* < 0.0001) (Figure 3). Upon visual inspection, relatively more non-survivors (Figure 3, Panel A) are located in the bottom right quadrant of the graph (patients from category 2 with the most severe acidaemia and hyperlactatemia) than in the remaining quadrants. The upper left quadrant contains the fewest non-survivors (patients from category 0). When severe renal failure was considered, patients from category 2 did not present eGFR < 30 mL/min/1.73 m^2^ more often than patients from other categories.

Associations between creatinine concentration or eGFR and BE, aBE, and lactate concentration are shown in Table 4. Lactate concentration was found not to correlate with either creatinine or eGFR, whereas BE and aBE were significant, although weakly related. Worsening of renal function was associated with deteriorating acidosis, regardless of whether BE or aBE was considered.

## 4. Discussion

We report that lactate concentration, standard BE, and alactic BE measured on admission to the ICU in patients with shock can have prognostic values for determining 28-day ICU mortality. The prognostic value of these parameters is significant regardless of the effects of age, gender, type of shock, and presence of eGFR < 30 mL/min/1.73 m^2^. Moreover, the prognostic value of standard BE and lactate concentration are independent and additive, with their combined values allowing for the classification of shock patients into low, moderate, and high-risk categories of mortality.

A direct comparison of our findings with other studies is problematic. Lactate and BE are routinely measured in critically ill patients suffering from various diseases, and both serve as mortality predictors in ICU (for example, in post-cardiac surgery patients or those with cardiogenic shock, ruptured abdominal aortic aneurysms, or trauma) [4,31,32,33,34,35,36,37,38,39,40,41]. Whether BE or lactate concentrations serve as better predictors of mortality in critical care patients has not yet been determined [4,33,34,35,36], with no studies reporting with certainty whether these predictors are both independent and/or additive.

All studies focus on critically ill patients. However, substantial differences in clinical features still exist (e.g., the cause and severity of specific clinical conditions, treatment methods, and length of stay or follow-up).

Smith et al. [42] found that both BE and lactate correlated with mortality for 148 patients admitted to ICU for different reasons. However, upon admission, the SOFA score for all patients was ≤5, with approximately 70% needing mechanical ventilation and 35% requiring an intravenous infusion of catecholamines. In contrast, the median SOFA for patients enrolled in this study was 13 points, with all patients requiring mechanical ventilation and infusion of at least norepinephrine.

Similarly, Schork et al. reported a relationship between lactate, BE, and pH and mortality in 4067 intensive care unit patients [40]. The predicted mortality by SAPS II was 29%, and only 47% of them required mechanical ventilation. For admission to ICU, the cut-off values for mortality were 2.1 mmol/L for lactate and −3.8 mmol/L for BE [40]. Compared with that study, all our patients were mechanically ventilated, their predicted mortality assessed by the SOFA score was around 55%, and the cut-offs for lactate and BE were 4.5 mmol/L and −9.5 mmol/L, respectively.

Wernly et al. [43] studied approximately 5600 septic patients. They found simultaneous acidosis and hyperlactataemia to be stronger predictors of mortality than either acidosis or lactate concentration (>2.3 mmol/L) alone. Similarly to our study, all patients were categorised into three groups according to acidosis with BE ≤ 6 and hyperlactataemia with lactate concentration >2.3 mmol/L. The cut-off values of BE and lactate concentrations for a higher risk of 28-day ICU mortality suggest that the patients in our study had even more advanced acidaemia (−6 vs. <−9.5 mmol/L for BE) and hyperlactataemia (>2.3 vs. >4.5 mmol/L for lactates) than those in Wernly et al.’s study [43]. Additionally, the patients in our study suffered from more severe shock, with all (vs. 76% for Wernly et al. [43]) requiring intubation with mechanical ventilation and intravenous infusion of norepinephrine, with an average SOFA score approximately six points higher (12.7 ± 3.25 vs. 7 ± 4).

It is also worth noting that some parameters, e.g., BE, are computed using different formulas in various studies. For example, Wernly et al. [43] used the Van Slyke equation for BE computation [44], whereas Smith et al. [42] used BE measured directly by a standard ward-based arterial blood gas analyser (Radio-meter ABL system 625/620), providing no formula. We apply the same formula in our practice as Gattinoni [28]. For these reasons, any comparison between various studies becomes difficult.

Both BE and lactate sum up to aBE, which, according to Gattinoni et al. [28], is a marker of a possible accumulation of plasma acids excreted by kidneys. It thus indirectly reflects the renal regulation of acid–base balance. Gattinoni et al. [28] have postulated that aBE measurements should improve the management of critically ill patients. Wernly et al. [43] were the first to demonstrate that aBE has any, although weak, predictive value for ICU mortality in a large group of 5586 septic patients (AUC 0.56, 95% CI 0.54–0.58). We expand upon this, demonstrating that admission aBE may predict the risk of 28-day ICU mortality in shock patients regardless of the impact of several other covariates, including the presence of severe renal failure.

Gattinoni et al. [28] postulated that hyperlactataemia accompanies acidaemia if renal function deteriorates. In our study, Cox regression models adjusted to the presence of eGFR < 30 mL/min/1.73 m^2^ confirmed that BE, aBE, and lactate concentration were all predictors of death in patients with and without severely impaired renal function. Table 4 shows that, in contrast to BE and aBE, lactate concentration is not correlated with indices of renal function. It can be seen from Figure 3 that lactate and BE are negatively related. Four quadrants defined by BE and lactate cut-offs illustrate how survivors and non-survivors separate into groups of varying mortality rates. It also aids in visualising how a combination of BE and lactates further stratifies patients in shock into categories of lowest, moderate, and highest mortality. This figure independently confirms the results of Cox regression models and survival curves, separating all patients into three categories according to BE and lactate concentration values. To our knowledge, no other study has reported similar findings. Figure 3 also demonstrates that individuals with severe renal failure are relatively equally scattered across the lactate concentration and BE correlogram.

Similarly to other studies, our investigation confirms that patients in shock with severe acidosis (BE < −9.5 mmol/L) or hyperlactataemia (lactate > 4.5 mmol/L) upon admission are at increased risk of ICU mortality. Novel findings of our study are as follows: firstly, alactic BE has not been shown before to predict mortality in shock patients. We demonstrate that alactic BE measured upon admission (<–3.63 mmol/L) and indicating the presence of severe acidosis originating from non-lactic organic acids may also predict ICU mortality. Second, the predictive value of lactates and BE is independent and additive. When both are combined, three groups of shock patients can be separated, low, medium, and high risk. The coexistence of both severe acidosis and hyperlactatemia (category 2) appeared to be the most vital risk factor for premature death in ICU shock patients. Finally, the cut-off values for BE and lactates indicate that analysed high-risk patients present more profound acidosis and hyperlactataemia than in other studies suggesting that people undergoing more severe acidosis are treatable and can be rescued.

Targeting such metabolic abnormalities is complex, and it is unknown whether such patients will benefit from even more attention and even more aggressive therapy, but it is always worth trying to intensify our management. The risk stratification of shock patients into low, medium, and high-risk categories based on the admission acid-base analysis, allows for a very early selection of patients with the highest risk of dying with the most severe acidosis and hyperlactataemia. Our finding appears to yield interesting clinical and prognostic information and might be used for future planning of a prospective study on such individuals. We also underline that alactic BE may provide predictive information for 28-day mortality. However, the combination of lactates and BE appears to outperform the prognostic information derived from either lactate alone, BE alone, or alactic BE.

### Limitations of This Study

It was a single-centre study involving a relatively small number of patients. However, many other similar studies are also single-centred. Included patients were admitted to the ICU at various stages of their acute critical condition, a shock of mixed aetiology. Some were admitted directly from their home due to sudden cardiac arrest, cardiovascular collapse, or out-of-hospital environment after accidents. Other patients were already hospitalised at other departments, e.g., surgery, cardiology, or neurology. Nevertheless, all patients were enrolled consecutively without complex inclusion or exclusion criteria, so the distribution of different forms of shock and the severity of clinical conditions reflect daily ICU practice. However, the rate of different shocks and patient management may not be the same in various hospitals and ICU units. Compared to other studies, a new limitation was introduced by cut-off values for BE and lactates to predict death during hospitalisation in ICU. It appears to be a consequence of distinct clinical features and/or various formulas used for BE calculation. We have deliberately applied the same formula as Gattinoni et al. [28] to overcome this issue partially. Even though the reported cut-offs for BE and lactate concentration in our subjects suggest that they were undergoing more severe acidosis than subjects in other studies [7,8,13,33,40,42,43,45]. It may also suggest that their progression in intensive care shifts the border of severe acidosis from a morbid to treatable zone, giving more high-risk patients hope for survival.

## 5. Conclusions

We have demonstrated that aBE <− 3.63 mmol/L may be associated with an increased risk of 28-day all-cause mortality. Additionally, the predictive value of BE and lactate concentration, which contributes to aBE, is independent and additive. Finally, the combination of BE < −9.5 mmol/L and lactate > 4.5 mmol/L better selects patients at the highest risk of death than BE or lactate concentration alone, or aBE. Notably, the predictive value of these metabolic indices is significant regardless of the influence of patients’ age, gender, a form of shock, and the presence of severe renal failure.

## Figures and Tables

**Figure 1 jcm-11-06125-f001:**
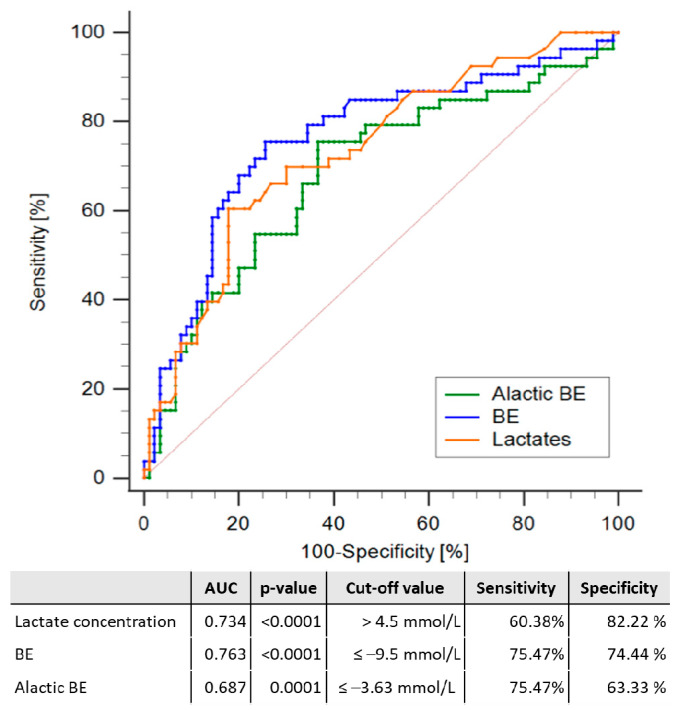
Results of the Receiver Operator Curve characteristics analysis for the prediction of 28-day mortality in shock patients hospitalised in intensive care by aBE (green line), BE (blue line), and lactate concentration (orange line). Median values of areas under the curve (AUC), optimal cut-offs, specificity and sensitivity of BE, lactic concentration, and aBE are shown underneath the graph. Abbreviations: AUC—area under the curve; BE—base excess.

**Figure 2 jcm-11-06125-f002:**
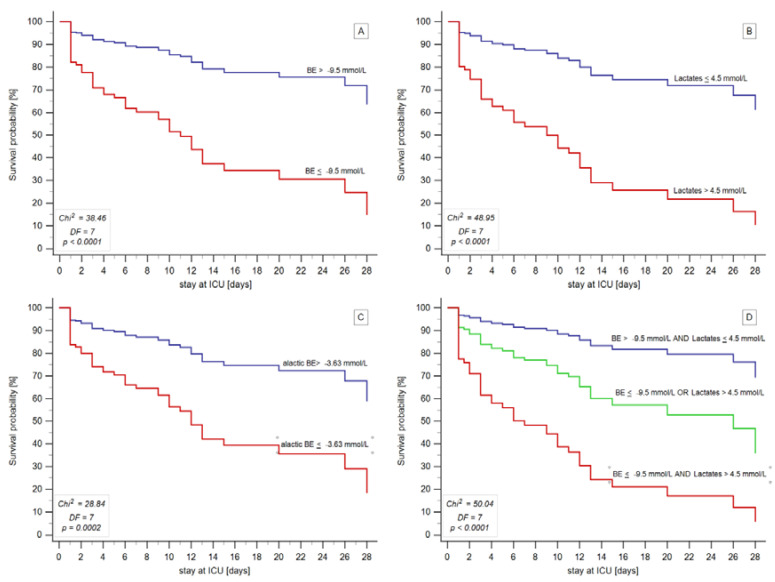
Comparison of survival curves modelled by the Cox proportional hazard regressions for the 28-day ICU mortality in shock patients, stratified according to the admission values of BE (Panel (**A**)), lactate concentration (Panel (**B**)), aBE (Panel (**C**)), and the combination of BE and lactate concentration (Panel (**D**)) based on cut-off values defined by the ROC analysis. Red lines correspond to patients with the highest risk, whereas blue lines correspond to those with the lowest risk. The green line in Panel (**D**) corresponds to the middle-risk group (category 1). Values of chi^2^, degrees of freedom (DF), and *p* for the models are shown in the bottom-left corners of each panel. Abbreviations: BE—base excess; aBE—alactic base excess, DF—degrees of freedom; ICU—intensive care unit.

**Figure 3 jcm-11-06125-f003:**
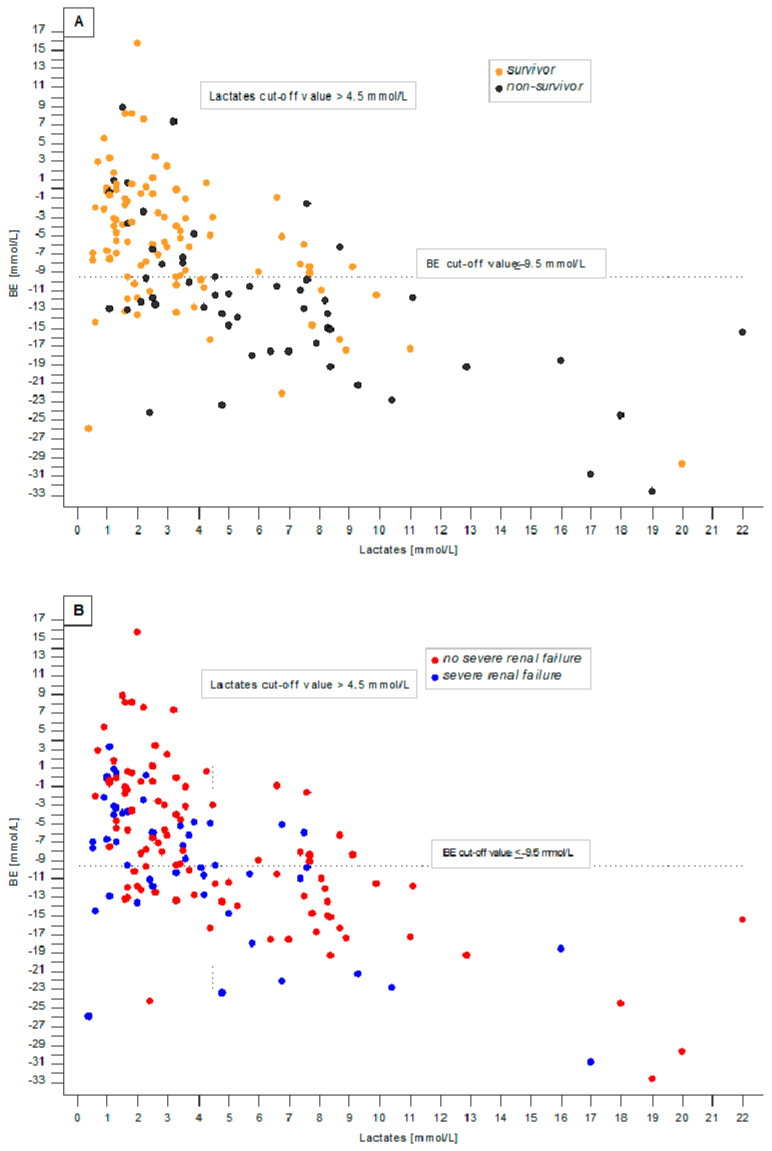
Association between BE value and lactate concentration in all patients, with a horizontal line for the BE cut-off and a vertical line for the lactates cut-off. In this way, four quadrants are presented. Patients in category 0 are in the upper left quadrant, and those from category 2 are in the bottom right quadrant. Individuals from category 1 are either in the upper right or bottom left quadrants. The same association is shown in both panels (**A**,**B**). However, in panel (**A**), patients are marked according to their ICU survival status at the end of the 28-day follow-up, whereas in panel (**B**), patients are marked according to the presence of severe renal failure, i.e., eGFR < 30 mL/min/1.73 m^2^.

**Table 1 jcm-11-06125-t001:** Summary of comparisons of clinical characteristics in a group of patients with shock divided into survivors and non-survivors of the 28-day stay at ICU.

Qualitative Data											
	Survivors					Non-Survivors					^#^ *p* Value
Parameter	N		%			N		%			
Number of patients	90		63			53		37			
Men	65		72.2			30		56.6			0.0677
Cardiogenic shock	18		20.0			18		34.0			0.0743
Hypovolemic shock	23		25.6			5		9.4			0.0278
Septic shock	49		54.4			30		56.6			0.8626
Other shocks	4		4.4			3		5.7			0.7102
Additional catecholamine	23		25.6			27		50.9			0.0034
eGFR < 30 mL/min/1.73 m^2^	29		32.2			20		37.7			0.5849
SaO_2_ < 94%	28		31.1			27		50.9			0.0216
standard HCO_3_^−^ < 22 mmol/L	49		54.4			46		86.8			<0.0001
pH < 7.35	61		67.8			45		84.9			0.0295
BE < −3 mmol/L	60		66.7			46		86.8			0.0098
Lactate concentration > 2 mmol/L	51		56.7			46		86.8			0.0002
**Continuous and Discrete Data**											
	**Survivors**					**Non-Survivors**					
**Parameter**	**Mean**	**SD**	**Median**	**25 P.**	**75 P.**	**Mean**	**SD**	**Median**	**25 P.**	**75 P.**	***p* Value**
Age (years)	58.42	14.54	61	47	69	65.51	11.85	67	59	72	0.0031
APACHE II	21.91	7.49	22	16.25	27	31.55	7.38	32	27	38	<0.0001 *
SOFA	11.67	3.09	12	10	14	14.45	2.76	15	12.75	16	<0.0001 *
Length of ICU stay (days)	9.57	8	6	4	14	5.78	6.41	3	1	10	0.6427 *
HR (beats/min)	108.73	17.99	110	95	120	111.04	22.94	115	100	120	0.3364 *
Systolic BP (mmHg)	123.69	29.60	120.00	105.00	145.00	105.111	31.8764	109.000	82.75	122.00	0.0236 *
Diastolic BP (mmHg)	66.79	15.64	67.00	58.00	80.00	53.556	17.3700	56.000	40.50	64.25	0.0026 *
Creatinine (mg/dL)	2.05	1.66	1.44	0.98	2.7	2.3	1.64	1.68	1.14	2.82	0.162 *
pH	7.29	0.13	7.3	7.23	7.38	7.17	0.16	7.2	7.12	7.29	<0.0001
Standard HCO_3_ (mmol/L)	21	5.46	20.95	17.7	24.2	16.1	5.79	15.7	13.28	18.25	<0.0001
Standard BE (mmol/L)	−5.65	7.17	−5.46	−9.53	−0.93	−12.37	8.07	−12.26	−16.98	−9.15	<0.0001
Lactate level (mmol/L)	3.36	3.03	2.45	1.3	3.9	6.43	4.85	5	2.58	8.3	<0.0001 *
aBE (mmol/L)	−2.29	6.13	−1.93	−6.24	1.44	−5.93	6.39	−6.46	−10.19	−3.36	0.001

^#^ Fisher exact test for binomial data; * Mann–Whitney test. Abbreviations: aBE—alactic base excess; APACHE II—the Acute Physiology and Chronic Health Evaluation II; BE—base excess; BP—blood pressure; HCO_3_—bicarbonate concentration; HR—heart rate; ICU—intensive care unit; P.—percentile; pH—the power of hydrogen; SOFA—Sequential Organ Failure Assessment.

**Table 2 jcm-11-06125-t002:** The mortality rate in specific subgroups of patients stratified to cut-off values of BE; lactate concentration; alactic BE; or categories 0, 1, and 2 based on combined cut-offs of BE and lactate concentration.

Stratifying Variable	Non-Survivors	Mortality Rate (%)
BE > −9.5 mmol/L	13	16.25
BE < −9.5 mmol/L	40	63.49
Lactate concentration < 4.5 mmol/L	21	22.11
Lactate concentration > 4.5 mmol/L	32	66.67
aBE > −3.63 mmol/L	13	18.57
aBE < −3.63 mmol/L	40	54.79
Category 0 (BE > −9.5 mmol/L and lactates < 4.5 mmol/L)	11	15.71
Category 1 (either BE < −9.5 mmol/L or lactates > 4.5 mmol/L)	12	21.81
Category 2 (both BE > −9.5 mmol/L and lactates < 4.5 mmol/L)	30	78.94

Abbreviations: aBE—alactic base excess; BE—base excess.

**Table 3 jcm-11-06125-t003:** Results of unadjusted and adjusted Cox proportional hazards regression models for the 28-day ICU mortality in patients with shock. Adjustments were made according to patients’ age, gender, presence of severe renal failure (eGFR < 30 mL/min/1.73 m^2^), and type of shock.

	Unadjusted Model			Adjusted Model		
Stratifying Variable	HR	95% CI	*p* Value	HR	95% CI	*p* Value
BE < −9.5 mmol/L	4.26	2.27–7.98	<0.0001	4.22	2.21–8.05	<0.0001
Lactate concentration > 4.5 mmol/L	3.58	2.05–6.23	<0.0001	4.62	2.56–8.33	<0.0001
aBE < −3.63 mmol/L	3.09	1.65–5.78	0.0004	3.19	1.62–6.27	0.0008
Change of one category	2.68	1.88–3.84	<0.0001	2.78	1.94–4.01	<0.0001

Abbreviations: aBE—alactic base excess; BE—base excess; CI—confidence interval; HR—hazard ratio.

**Table 4 jcm-11-06125-t004:** Spearman correlation between renal function indices and BE, aBE, and lactate concentration.

	Creatinine Concentration		eGFR	
Parameters	Rho	*p* Value	rho	*p* Value
BE	−0.29	0.0004	0.31	0.0002
Lactate concentration	0.02	0.7952	−0.03	0.7216
aBE	−0.36	0.0000	0.37	0.0000

Abbreviations: BE—base excess; aBE—alactic BE; eGFR—estimated glomerular filtration rate; rho—coefficient of nonparametric Spearman correlation.

## Data Availability

The datasets generated and/or analysed for this study are currently not publicly available due to their in other analyses. Selected data, however, are available from the corresponding author upon request.

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
