# Peer review of "Admission Lactate Concentration, Base Excess, and Alactic Base Excess Predict the 28-Day Inward Mortality in Shock Patients"

_jcm, 2022, doi:10.3390/jcm11206125_

Round 1
Reviewer 1 Report
Dear authors,
thank you for giving me the opportunity to evaluate this manuscript. I reviewed the work on the basis of the manuscript and Tab. 1-3, Figure 1-2 and the supplementary. The major objective of this retrospective, single-center study was to evaluate the influence of Base excess(BE), alactic BE and lactate concentration on the prediction of the 28-day mortality. Thus, this manuscript is divided in to introduction, materials and methods, results and discussion. nine section including introduction and conclusion. Piotr Smzszkiewicz and colleges have analyzed their research question using Receiver Operating Curve (ROC) charateristics and Cox proportional hazard regression models. The main finding was that the predictive values of BE and lactate were independent and additive. I have major comments regarding this manuscript.
Major comments:
Material and Methods
l.79-84. What do you mean with sensitive patient data? How did you make them anonymous?
In this sentence also a dot is missing. Who approved the study protocol?
Results
l.96-99. The description of qualitative data is not necessary from my point of view.
In Table 1 the number of all patients, of all survivors and non-suvirvors a messing. The general format is hard to read and understand. Please summarize your results in a better way, thus the table can be printed vertical on the page. In terms of multiple testing errors please do not test every parameter on significance differences. Please choose only mean or median and the corresponding depending dispersion measure based on the distribution of the parameters and logical value of the parameter.
In Table 3 it is not recognizable which column belongs to unadjusted and adjusted Cox proportional hazards regression model. Was there also a different between the category change from 0 to 1 and 1 to 2, respectively? Please describe in more detail.
l. 210 “and3.02…” a space is missing.
Discussion
l. 235-241 What a the potential reasons why your cut-off values are different to named study?
l. 273-282 In this part you discuss results that were not presented in the result part. If you think these are important results than mention them also in the result part. Especially, if you say these findings are reported for the first time.
l. 301 Instead of centers you probably mean departments. Please check it.
General
From my point of view analyzing aBE is specialty of your work. Thus, put it the foreground of your work.
Thank you.
Author Response
We want to thank all Reviewers for their work, suggestions, goodwill, intentions to help us, and time spent on our manuscript. Below, we reply in a step-by-step way to all questions and issues. We corrected all that was necessary. In cases where we have not agreed, we have replied and explained our position carefully.
Review 1
Dear Madam/Sir
Thank you for all your effort and suggestions. Please find our replies below.
Major comments:
Material and Methods
l.79-84. What do you mean with sensitive patient data? How did you make them anonymous?
Reply - For this study, we collected clinical data in our internal database. However, we have access to all patients' data under our care, including sensitive ones like family names, home address details, telephone number(s), email(s) etc. All patients were instantly given unique codes to protect their sensitive data against potential leakage or circulation. Such procedures are obligatory in Poland. We have clarified it and made necessary amendments to the text:
All patients were given unique codes to keep their anonymity in the database. No names or contact details were circulating among researchers involved in the study. In this way, patients' anonymity was preserved.
In this sentence also a dot is missing.
Reply – it has been corrected.
Who approved the study protocol?
Reply - The Bioethics Committee approved the study protocol at the Poznan University of Medical Sciences. This information is provided in the Methods section, and it goes as follows:
All patients recruited for this study were unconscious at the time of enrollment. Thus, the Bioethics Committee at the Poznan University of Medical Sciences (www.bioetyka.ump.edu.pl) waived the requirement to obtain informed consent in Poland and approved the study protocol.
Results
l.96-99. The description of qualitative data is not necessary from my point of view.
Reply - Thank you for the comment. This part has been removed.
In Table 1 the number of all patients, of all survivors and non-suvirvors a messing.
Reply - Now we have added the proper number for all patients, survivors and non-survivors.
The general format is hard to read and understand. Please summarize your results in a better way, thus the table can be printed vertical on the page. In terms of multiple testing errors please do not test every parameter on significance differences. Please choose only mean or median and the corresponding depending dispersion measure based on the distribution of the parameters and logical value of the parameter.
Reply - We have two types of tests because of the various distributions: the t-test and the Mann-Whitney test. The former, in its algorithm, uses both the mean and the standard deviation (in the form of pooled SD). A statistically savvy reader can glean the clinical importance of the carried test by looking at the means, standard deviations, and group sizes. On the other hand, the Man-Whitney uses the median to rank individual measurements and build the pivotal variable - the median is a must in reporting results analysed by this test. Again, by looking at the median and IQR it is possible to derive information on whether the unimodality assumption necessary for the Man-Whitney test holds. So IQR is also necessary for this test. Therefore, we prefer to leave all data in this table as they are.
In Table 3 it is not recognizable which column belongs to unadjusted and adjusted Cox proportional hazards regression model.
Reply - Thank you for pointing out this serious problem. It has been corrected.
Was there also a different between the category change from 0 to 1 and 1 to 2, respectively?
Reply - Yes, indeed there was. The Cox proportional hazards regression models "balance" the hazards between different categories, and the results should be read as how much the hazard ratio changes per a single step in a specific category. In other words, HR values show the median change of the hazard per a single change in a category. The same hazard is for the change between 0 and 1, and then between 1 and 2.
- 210 "and3.02…" a space is missing.
Reply - It has been corrected.
Discussion
- 235-241 What a the potential reasons why your cut-off values are different to named study?
Reply - Our patients at admission probably had more severe forms of shock and acid-base abnormalities than in other studies (all patients were on mechanical ventilation and infusion of at least one catecholamine, and they had higher SOFA scores than patients in other studies). So, different clinical characteristics might explain the differences in the cuf-offs.
Thanks to the progress in the medical care of shock patients, we can now treat people with more advanced acidosis and hyperlactatemia, and they survive! The mortality rate in our study was 37%, which for such a group of shock patients is much lower than reported in older studies. It might also influence the ROC analysis and selection of the cut-off thresholds.
- 273-282 In this part you discuss results that were not presented in the result part. If you think these are important results than mention them also in the result part. Especially, if you say these findings are reported for the first time.
Reply - Thank you for this suggestion. We have moved the data from the Supplementary materials to the main text.
- 301 Instead of centers you probably mean departments. Please check it.
Reply - Thank you for spotting the problem. We meant general differences between the different hospitals, ICU units, countries etc., not only departments within the same hospital. At our hospital, which is localized in four different places, we have a couple of ICUs, some of which are dedicated to specific patients, like after cardiosurgery, neurosurgery, otolaryngology, general surgery, and internal medicine, or intensive cardiac therapy. To avoid confusion, we have modified the sentence.
Nevertheless, all patients were enrolled consecutively without complex inclusion or exclusion criteria, so the distribution of different forms of shock and the severity of clinical conditions reflect daily ICU practice. However, the rate of different shocks and ways of patient management may not be the same in various hospitals and ICU units.
General
From my point of view analyzing aBE is specialty of your work. Thus, put it the foreground of your work.
Reply - We are grateful for the appreciation of our work. A considerable part of our study is indeed on aBE, and it appears that we are the first to show its predictive value in shock patients. We have modified the discussion to underline it better.
Thank you again for your review.
Reviewer 2 Report
Introduction: Appears complete and thorough.
METHODS:
I have some questions about the inclusion criteria that should be clarified. It states that 143 consecutive patients requiring norepinephrine were enrolled; however, it appears that all patients also required mechanical ventilation. Was there also a requirement for mechanical ventilation?
Also ,did data collection begin on the date of ICU admission or the time at which they met the inclusion requirement? Since there is only one point given (BE, aBE and lactate), it is unclear if these were the single worst values during ICU stay, the first value in the ICU, the first after meeting criteria, etc. The authors do say that data were collected 'at enrollment.' How much time was between the lactate and the ABG measurement from which BE was recorded? Did the investigators use the most temporally near or the value that would give the best (or worst) result in a specific period of time (for example, on the date of enrollment)? Also, it might be helpful to understand if patients were admitted to the ICU in this condition or if some may have been in the ICU and deteriorated to multisystem organ failure (as the authors note, at least 3 organ failures based on mentation, ventilation and vasopressor requirement).
In addition, specific data regarding adherence to guideline-based treatment would be helpful, such as tidal volumes used and timing / appropriateness of antibiotics, as theses parameters have association with mortality in patients with shock (in the case of antibiotics, septic shock).
RESULTS:
- Again, result interpretation would be facilitated by knowing the details of data collection as noted above; specifically, how the BE, lactate and aBE were recorded in the case of multiple values in the date of enrollment.
- ICU length of stay when looking at survivors vs nonsurvivors is not a meaningful parameter. Rather than display the data by dependent variable, it may be more meaningful to display the data by independent variable (BE, aBE and lactate). The KME panels are an example of this, and seem to be more meaningful.
- The overall finding of the paper is expected. One would anticipate failure of the metabolic process (as noted by worse BE, aBE and lactate) as patients are in the process of dying with cardiopulmonary failure (and overall multisystem organ failure). Indeed, the finding of elevated lactate correlating with mortality is not novel. Perhaps it being additive to the other parameters is novel.
DISCUSSION:
Overall the discussion is thorough. It would be preferable that the authors include a brief discussion of how their findings might influence clinical care. Some of this might require further exploration of limitations of the study or clarification in the methods and data provided. For example: how long were patients in the hospital and / or ICU prior to enrollment? How much time lapsed between the measurement and the outcome (death or transfer from the ICU)? How many of the patients were enrolled after cardiac arrest?
GENERAL COMMENTS:
Clarification of the above concerns is important. Waiting until a patient requires both vasopressors and mechanical ventilation to enroll patients in a retrospective study will pre-select a population with very high mortality, and at this point the outcome may not be modifiable - thus explaining some of the findings and bringing question into the clinical significance of this finding. The goal of most critical care physicians is to intervene prior to multisystem organ failure and thus prevent the organ failures leading to mortality. Rather than looking at the characteristics of those who survived vs those who did not, it may be more meaningful to look at the outcomes of those who had worse BE, aBE and lactate; that might be more influential on clinical decision making.
Author Response
We want to thank all Reviewers for their work, suggestions, goodwill, intentions to help us, and time spent on our manuscript. Below, we reply in a step-by-step way to all questions and issues. We corrected all that was necessary. In cases where we have not agreed, we have replied and explained our position carefully.
Review 2
Dear Madam/Sir
Thank you for all your effort and suggestions. Please find our replies below.
METHODS:
I have some questions about the inclusion criteria that should be clarified. It states that 143 consecutive patients requiring norepinephrine were enrolled; however, it appears that all patients also required mechanical ventilation. Was there also a requirement for mechanical ventilation?
Reply - Thank you for spotting this issue which might be unclear. All patients indeed required mechanical ventilation. We have included it in the modified inclusion criteria as it is crucial information that better reflects the severity of the clinical condition of our patients.
Also ,did data collection begin on the date of ICU admission or the time at which they met the inclusion requirement Since there is only one point given (BE, aBE and lactate), it is unclear if these were the single worst values during ICU stay, the first value in the ICU, the first after meeting criteria, etc. The authors do say that data were collected 'at enrollment.' How much time was between the lactate and the ABG measurement from which BE was recorded? Did the investigators use the most temporally near or the value that would give the best (or worst) result in a specific period of time (for example, on the date of enrollment)? ?
Reply - Routinely, patients admitted to our ICU have collected blood samples as soon as possible. Usually, it is within the first 5-10 minutes after admission. It is what we mean by the admission BE, aBE, lactate, and other biochemical parameters. All patients admitted to ICU fulfilling the inclusion criteria were enrolled in the study, and all patients were consecutive. The enrollment initiated no special actions or changes in the therapy. Whatever was done is routinely performed at our ICU. In other words, enrollment meant that the data routinely collected of a particular patient would be used. For this study, we used only data from the admission, as explained above.
Also, it might be helpful to understand if patients were admitted to the ICU in this condition or if some may have been in the ICU and deteriorated to multisystem organ failure (as the authors note, at least 3 organ failures based on mentation, ventilation and vasopressor requirement).
Reply – It is an important issue that we must clarify. Thank you for bringing it to our attention. All patients enrolled in this study were new patients not hospitalized before at any other ICU in our or another hospital for the current clinical state. In contrast, patients hospitalized anywhere at ICU for a different reason in the past (like a year or so) and then discharged home, were included. We were interested in new unconscious patients who developed any shock and required at least norepinephrine and mechanical ventilation. We have not included patients transferred from other ICUs.
In addition, specific data regarding adherence to guideline-based treatment would be helpful, such as tidal volumes used and timing / appropriateness of antibiotics, as theses parameters have association with mortality in patients with shock (in the case of antibiotics, septic shock).
Reply - All patients were treated according to standard procedures in mechanical ventilation, shock management, hemodynamic monitoring, and appropriate use of antibiotics.
RESULTS:
- Again, result interpretation would be facilitated by knowing the details of data collection as noted above; specifically, how the BE, lactate and aBE were recorded in the case of multiple values in the date of enrollment.
Reply - All blood samples were collected within 5-10 minutes after admission to our ICU. The first blood gases, acid-base, and other biochemical parameters were measured at the bedside with the (Radiometer ABL90 Flex Plus) from the arterial blood samples. This device is in our ICU; thus, all results are available instantly and on-demand at any time.
- ICU length of stay when looking at survivors vs nonsurvivors is not a meaningful parameter. Rather than display the data by dependent variable, it may be more meaningful to display the data by independent variable (BE, aBE and lactate). The KME panels are an example of this, and seem to be more meaningful.
Reply - We have stratified our patients according to the estimated thresholds of BE, lactates, aBE, or the combination of BE and lactates into separate groups. The patients' stratification was based on the dichotomized values of independent parameters like BE, lactates, and aBE. In survival analysis, for example, Kaplan-Meier curves or Cox proportional hazard regression models (as in our case), time is an independent parameter. We studied the 28-day mortality based on the length of the ICU stay. For this reason, the length of ICU stay in days reflects the survival time.
- The overall finding of the paper is expected. One would anticipate failure of the metabolic process (as noted by worse BE, aBE and lactate) as patients are in the process of dying with cardiopulmonary failure (and overall multisystem organ failure). Indeed, the finding of elevated lactate correlating with mortality is not novel. Perhaps it being additive to the other parameters is novel.
Reply - Our study has two parts, confirmatory in which we show that both lactates and BE, separately, may predict mortality. There is also another part that seems to be novel which includes two findings. First, alactic BE has not been shown before to predict mortality in shock patients. We demonstrate that alactic BE measured upon admission (< -3.63 mmol/L) and indicating the presence of severe acidosis originating from non-lactic organic acids may also predict ICU mortality. Second, the predictive value of lactates and BE are independent and additive. When both are combined, three groups of shock patients can be separated, lower, medium, and high risk. The coexistence of both severe acidosis and hyperlactataemia (category 2) is the most vital risk factor for premature death in ICU in shock patients. Finally, the cut-off values for BE and lactates indicate that high-risk patients present more profound acidosis and hyperlactataemia than in other studies suggesting that people undergoing more severe acidosis are treatable and can be rescued. It may also suggest that their progression in intensive care shifts the border of severe acidosis from a morbid to treatable zone, giving more high-risk patients hope for survival.
DISCUSSION:
Overall the discussion is thorough. It would be preferable that the authors include a brief discussion of how their findings might influence clinical care. Some of this might require further exploration of limitations of the study or clarification in the methods and data provided. For example: how long were patients in the hospital and / or ICU prior to enrollment?
Reply - We appreciate these comments and suggestions on how to improve our discussion.
Our conclusions can be limited to the enrolled patients, i.e., those who fulfilled the inclusion criteria. As we wanted to be as general as possible, we used three common criteria. They included unconscious patients with (1) newly diagnosed shock who required two interventions: (2) at least norepinephrine infusion; and (3) mechanical ventilation. These criteria cover many ICU patients and reflect the everyday practice in our department.
Our findings suggest a couple of things. First, as our patients appear to have more severe acidosis and hyperlactatemia than in other studies, we show that many of them are treatable. Second, it is possible to stratify patients into low, medium, and high-risk categories. As this stratification is based on the admission acid-base analysis, a very early selection of patients at the highest risk is possible. It is unknown whether such patients will benefit from more attention and even more aggressive therapy, but it is always worth trying to intensify our management. Our findings might be used for future planning of a prospective study on such individuals. Additionally, we underly that alactic BE also provides predictive information. However, the combination of lactates and BE appears to outperform the prognostic information derived from either lactate alone, BE alone or alactic BE.
How much time lapsed between the measurement and the outcome (death or transfer from the ICU)?
Reply - As presented in Table 1, the median length of stay for the whole group was five days, for non-survivors three days, and those who survived six days. It suggests that those who died were in more serious conditions and their life expectancy was shorter. On the other hand, patients who survived required more time to become eligible for discharge, usually to other non-ICU departments.
How many of the patients were enrolled after cardiac arrest?
Reply - Fourteen patients out of 143 (9.8%) were admitted after successful CPR of cardiac arrest, Their 28-day ICU mortality was 50%.
GENERAL COMMENTS:
Clarification of the above concerns is important. Waiting until a patient requires both vasopressors and mechanical ventilation to enroll patients in a retrospective study will pre-select a population with very high mortality, and at this point the outcome may not be modifiable - thus explaining some of the findings and bringing question into the clinical significance of this finding. The goal of most critical care physicians is to intervene prior to multisystem organ failure and thus prevent the organ failures leading to mortality. Rather than looking at the characteristics of those who survived vs those who did not, it may be more meaningful to look at the outcomes of those who had worse BE, aBE and lactate; that might be more influential on clinical decision making.
Reply - Thank you for these comments. We entirely agree with the Reviewer. However, there might be some misunderstanding. We did not wait to start medical treatment until the enrollment. It was the other way around. Patients were enrolled if they required not theoretically but practically, and in most, if not all, cases, they were already on the treatment with norepinephrine and mechanical ventilation. Another catecholamine (e.g., dobutamine, adrenaline) or vasopressin was added at different time points during the ICU stay, and they instead reflected more severe shock or resistance to the epinephrine alone treatment. Usually, patients who are admitted to our ICU are already intubated. Many are already on an IV epinephrine infusion, at least through the peripheral vein access.
Thank you again for the review.
Reviewer 3 Report
Interesting proof-of-concept paper. The retrospective study and the limited number of pts enrolled hamper its scientific relevance.
However, I suggest to stress the novelty in the discussion section and the future possible application.
Minor suggestions:
- add the number of pts in the table (top of the columns)
- has the number of organ dysfunction an impact on the prognosis and on the prognostic role of aBE?
Author Response
We want to thank all Reviewers for their work, suggestions, goodwill, intentions to help us, and time spent on our manuscript. Below, we reply in a step-by-step way to all questions and issues. We corrected all that was necessary. In cases where we have not agreed, we have replied and explained our position carefully.
Review 3
Dear Madam/Sir
Thank you for all your effort and suggestions. Please find our replies below.
Interesting proof-of-concept paper. The retrospective study and the limited number of pts enrolled hamper its scientific relevance.
However, I suggest to stress the novelty in the discussion section and the future possible application.
Reply - We are grateful for the appreciation of our study. We have extended the part of the novelty of our study and potential clinical application.
Our study has two parts, confirmatory in which we show that both lactates and BE, separately, may predict mortality. There is also another part that seems to be novel which includes two findings. First, alactic BE has not been shown before to predict mortality in shock patients. We demonstrate that alactic BE measured upon admission (< -3.63 mmol/L) and indicating the presence of severe acidosis originating from non-lactic organic acids may also predict ICU mortality. Second, the predictive value of lactates and BE are independent and additive. When both are combined, three groups of shock patients can be separated, lower, medium, and high risk. The coexistence of both severe acidosis and hyperlactataemia (category 2) is the most vital risk factor for premature death in ICU in shock patients. Finally, the cut-off values for BE and lactates indicate that high-risk patients present more profound acidosis and hyperlactataemia than in other studies suggesting that people undergoing more severe acidosis are treatable and can be rescued. It may also suggest that their progression in intensive care shifts the border of severe acidosis from a morbid to treatable zone, giving more high-risk patients hope for survival.
Minor suggestions:
- add the number of pts in the table (top of the columns) –
Reply – it has been corrected
- has the number of organ dysfunction an impact on the prognosis and on the prognostic role of aBE?
Reply: The SOFA score reflects the dysfunction of different organs, which was higher in the non-survivors than survivors. In the Cox proportional hazard model, regardless of whether the SOFA score was added to the model adjusted to age, gender, and type of shock with or without eGFR <30, the predictive value of aBE remained significant. We show the results below.
A Cox proportional-hazards regression model with eGFR < 30 but SOFA score
Covariate |
P value |
Odds ratio |
95% CI |
ABE <-3.63 = 1 |
0.0140 |
2.4225 |
1.1960 to 4.9069 |
female=0 |
0.7074 |
0.8946 |
0.5003 to 1.5999 |
Septic shock=1 |
0.1684 |
2.0233 |
0.7422 to 5.5153 |
Hypovolemic shock=1 |
0.0710 |
0.3050 |
0.0840 to 1.1071 |
Cardiogenic shock=1 |
0.3987 |
0.6472 |
0.2356 to 1.7778 |
eGFR30 < 30 = 1 |
0.7660 |
1.1081 |
0.5635 to 2.1792 |
Age per year |
0.0089 |
1.0328 |
1.0081 to 1.0580 |
SOFA score per point |
0.0039 |
1.1832 |
1.0556 to 1.3263 |
A Cox proportional-hazards regression model without eGFR < 30 but SOFA score
Covariate |
P value |
Odds ratio |
95% CI |
ABE <-3.63 = 1 |
0.0113 |
2.4620 |
1.2261 to 4.9436 |
female=0 |
0.7224 |
0.9003 |
0.5043 to 1.6072 |
Septic shock=1 |
0.1830 |
1.9355 |
0.7322 to 5.1163 |
Hypovolemic shock=1 |
0.0745 |
0.3062 |
0.0834 to 1.1240 |
Cardiogenic shock=1 |
0.3989 |
0.6438 |
0.2315 to 1.7908 |
Age per year |
0.0080 |
1.0330 |
1.0085 to 1.0582 |
SOFA score per point |
0.0021 |
1.1743 |
1.0603 to 1.3006 |
As renal failure is already counted in as one of the SOFA score elements, the model without eGFR < 30 but with SOFA is more suitable to answer this question. However, as we have preliminarily defined the adjusting covariates, we would prefer, if possible, to show the models which we have already presented in the manuscript.
Thank you for the review.
Round 2
Reviewer 1 Report
Dear authors,
thank you for the revision of your manuscript. But still, I want to point out the following.
1) Please focus on your main hypothesis. Thus, you may summarize the results in a more compact way. This means that you may consider to put the additional laboratory values into the supplementary material, especially those in table 1 and 2. Please check each value, if it is necessary to present them in the main manuscript (i.e. PaO2, CaO2, all vasopressors, etc.).
2) You state that the predictive value of your finding is independent of the shock form itself. This is an interesting result, which is worth to be presented in a more detailed way. You may also show the different ROC curves for each shock form as in figure 1. Please discuss, why the different shock forms may have no influence on your findings.
3) I still suggest not to test every single value on significance due to the risk of multiple testing and the associated increased rate of type I errors. Check if testing is necessary to underline your main hypothesis otherwise do not check for significance.
Thank you again.
Reviewer 2 Report
Overall, the manuscript is much improved. Thank you for the edits.
The content is interesting from the perspective of hypothesis generation. I would caution in using terms such as 'predictive' when performing retrospective data as you have in the conclusions. Even when consecutive subjects are included, there are many factors, seen and unseen, impacting the inclusion of patients in the retrospective cohort; indeed, these factors are why many retrospective observations do not prove true in prospective investigations. I would recommend softening this statement somewhat, eg, "may be predictive" or "could predict."
In any case, I still question the novelty of the observation. Similar outcomes would have been predicted with the APACHE-II and SOFA, with the SOFA score predominantly comprised of points derived from the inclusion criteria. In that case, I am not certain that the finding provides any clinically actionable data, as that degree of prognostication may have already been determined by the antecedent course, leading to the events of the inclusion criteria (intubation, vasoactive medication requirement, and altered mentation). In any case, it is all the more reason for further investigation.
